# Plasma-Induced Surface Modification of Sapphire and Its Influence on Graphene Grown by Plasma-Enhanced Chemical Vapour Deposition

**DOI:** 10.3390/nano13131952

**Published:** 2023-06-27

**Authors:** Miguel Sinusia Lozano, Ignacio Bernat-Montoya, Todora Ivanova Angelova, Alberto Boscá Mojena, Francisco J. Díaz-Fernández, Miroslavna Kovylina, Alejandro Martínez, Elena Pinilla Cienfuegos, Víctor J. Gómez

**Affiliations:** 1Nanophotonics Technology Center (NTC), Universitat Politècnica de València, 46022 Valencia, Spain; msinloz@ntc.upv.es (M.S.L.); igbermon@etsii.upv.es (I.B.-M.); tivanova@ntc.upv.es (T.I.A.); fradafer@ntc.upv.es (F.J.D.-F.); mikov@ntc.upv.es (M.K.); amartinez@ntc.upv.es (A.M.); epinilla@ntc.upv.es (E.P.C.); 2Institute of Optoelectronic Systems and Microtechnology (ISOM), Universidad Politécnica de Madrid, 28040 Madrid, Spain; alberto.bosca@upm.es

**Keywords:** surface plasma treatment, graphene, sapphire surface, plasma enhanced chemical vapor deposition

## Abstract

In this work, we study the influence of the different surface terminations of c-plane sapphire substrates on the synthesis of graphene via plasma-enhanced chemical vapor deposition. The different terminations of the sapphire surface are controlled by a plasma process. A design of experiments procedure was carried out to evaluate the major effects governing the plasma process of four different parameters: i.e., discharge power, time, pressure and gas employed. In the characterization of the substrate, two sapphire surface terminations were identified and characterized by means of contact angle measurements, being a hydrophilic (hydrophobic) surface and the fingerprint of an Al- (OH-) terminated surface, respectively. The defects within the synthesized graphene were analyzed by Raman spectroscopy. Notably, we found that the I_D_/I_G_ ratio decreases for graphene grown on OH-terminated surfaces. Furthermore, two different regimes related to the nature of graphene defects were identified and, depending on the sapphire terminated surface, are bound either to vacancy or boundary-like defects. Finally, studying the density of defects and the crystallite area, as well as their relationship with the sapphire surface termination, paves the way for increasing the crystallinity of the synthesized graphene.

## 1. Introduction

Graphene-based consumer electronics will not be affordable in large-scale fabrication until a reliable process of getting graphene on dielectrics without contamination and defects is attained [1,2]. Focusing on sapphire, which is a ubiquitous substrate in optoelectronic devices, the chemical vapor deposition (CVD) technique has been exploited as a possible fabrication route to obtain graphene-on-sapphire [2,3]. However, this technique presents several drawbacks, such as the high temperature (>900 °C) needed for the precursor to dissociate and nucleate the graphene on the non-catalytic surface [4,5,6,7]. On the other hand, plasma-enhanced CVD (PECVD) arises as a possible technological solution for reducing the demanding temperatures by aiding the process with the ignition of plasma [8,9]. The thin film formation can be simplified in nucleation, grain growth and coalescence steps. The plasma of the PECVD process promotes the decomposition of hydrocarbons, generating a large density of active radicals and species which adsorb in the substrate surface. During the nucleation process, these adatoms diffuse and create nucleation centers whose size increases with the subsequent adhesion of adatoms. Following this, the nucleation centers grow laterally, and their density rapidly saturates. Finally, they coalesce, forming a thin film. The competition between the nucleation and growth of graphene or its etching governs the PECVD synthesis of graphene. Either the growth or the etching state is determined by the process parameters, e.g., discharge power, temperature, hydrogen concentration or pressure [10,11]. Wei et al. identify the different regimes during the synthesis of graphene depending on the synthesis parameters (temperature, pressure and the H_2_:CH_4_ ratio) [10].

The properties of sapphire have been extensively studied for its use as a substrate in the synthesis of III-V semiconductors. There are different surface reconstructions and three stable terminations for c-plane sapphire, which strongly influence the properties of its surface [12]. For example, the surface termination determines the properties of III-V semiconductors grown on the sapphire surface [13,14,15]. Theoretical calculations have reported that the single Al-termination has the lowest surface energy, whereas the oxygen termination displays the largest [16,17,18]. On the other hand, the O-termination is stable only when hydrogen is present on the surface [19,20]. Upon heat treatment, the surface structure is reversible [21,22].

Considering the case of graphene synthesis on sapphire, several works report different nucleation behaviors depending on the crystal orientation or surface reconstruction of sapphire (Table 1). For example, because of the catalytic behavior of its surface, the nucleation density is largely increased when the synthesis is carried out on r-plane sapphire [23,24].

On the other hand, Mishra et al. reported a significant increase in the measured mobility of graphene synthesized by CVD on a c-plane sapphire, including an H_2_ thermal etching at 1180 °C [1]. They identified the oxygen-deficient (√31 × √31) R ± 9° reconstruction of sapphire via low energy electron diffraction (LEED) measurements and noted that the catalytic behavior of surface Al atoms increases the quality of their synthesized graphene, as compared with the pristine c-plane sapphire. The critical role of the surface termination of c-plane sapphire has also been demonstrated for the epitaxial growth of MoS_2_ on sapphire [26]. However, to the best of our knowledge, the influence of the different surface terminations of sapphire on the PECVD synthesis of graphene has not yet been addressed. In this work, the effect of the different sapphire surface terminations on graphene grown by PECVD is reported. Moreover, the sapphire surface terminations are controlled within the synthesis chamber by means of a plasma process, whose major parameters governing the process are the plasma power and its gas chemistry [27,28]. The induced modifications on the sapphire substrate have been characterized by means of contact angle and atomic force microscopy (AFM) measurements. Afterward, a defect analysis of the synthesized graphene was carried out using the Raman spectroscopy technique. Two different regimes of the defects are found (either vacancy or boundary-like defects), whereas the study of the crystallinity and the density of defects within the synthesized graphene yield valuable insight into promoting the larger size of the graphene grains on the sapphire substrate.

## 2. Materials and Methods

### 2.1. Substrate Cleaning

Pristine 4 inch c-plane sapphire wafers (PHOTONEXPORT Epi-Ready), covered with a protective PMMA resist to avoid splinters, were diced in 10 × 10 mm squares. The diced sapphire substrates were subjected to the following cleaning procedure: First, the PMMA resist was removed with an O_2_ plasma for 600 s at 400 W and 1.5 mbar (PVA TEPLA 200, PVA TePla AG). Afterward, the following 2-solvent method was employed: to remove any particles, they were rinsed below running deionized water for 30 s, blown dry in N_2_ and immersed in acetone for 300 s at room temperature (RT). Finally, the sapphire substrates were sonicated in isopropyl alcohol (IPA) for 300 s at RT, blown dry with N_2_ and introduced into the PECVD system (BlackMagic 6 inch, Aixtron Nanoinstruments Ltd., Herzogenrath, Germany).

### 2.2. Surface Modification of Sapphire: Design of Experiments

First, the top and bottom heaters of the synthesis chamber were heated up to 900 °C and 800 °C, respectively, with a heating rate of 200 °C/min in a pure Ar atmosphere at 4 mbar. To eliminate undesired temperature gradients within the synthesis chamber, the heaters remained at the target temperatures for 600 s at 800 °C. Afterward, the plasma process was initiated. The series of experiments were decided based on the table of signs or L_8_ orthogonal array (Table 2) to study the influence of 4 factors using a 2-level fractional factorial design 2^4−1^ [29]. The fractional factorial design, implemented with the Statgraphics Centurion XVIII software, screens major influences of the different factors under study along with the interactions between them. The 4 selected factors were: discharge power, process pressure, gas and process time. In our case, the process temperature was set to 800 °C after optimizing the synthesis of graphene on pristine c-plane sapphire surfaces (see Appendix A). The substrate was then cooled down at a rate of 15 °C/min, and the samples were unloaded at temperatures below 200 °C. Afterward, the sapphire substrates were characterized by means of water contact angle measurements in a ramé-hart automatized goniometer and 1 × 1 mm^2^ atomic force microscopy (AFM) measurements (MultiMode 8-HR, Bruker, Bremen, Germany) in tapping mode. Silicon AFM probes (VTESPA-300, Bruker; f_r_ = 300 kHz, K = 300 N/m) coated in the backside with reflective aluminum were employed. The AFM measurements were then analyzed using open-source Gwyiddion software.

### 2.3. PECVD Growth of Graphene

To minimize the variables induced by the PECVD system, the graphene synthesis was carried out within the same run in every sapphire sample. Additionally, a pristine c-plane sapphire substrate was introduced (reference sample). The samples were cleaned using the 2-solvent method described above and introduced into the PECVD chamber. Similar to the heating process explained before, the system stayed for 600 s in a pure Ar atmosphere with a process pressure of 4 mbar once the top and bottom heaters reached 900 °C and 800 °C, respectively, with a heating rate of 200 °C/min. Afterward, the supply of Ar was halted while the reactive gas (methane, CH_4_) was introduced with the following gas ratio 1:5:5; CH_4_, N_2_ and H_2,_ respectively, and the DC plasma turned on with a discharge power of 100 W. The growing process was set to 1200 s. The synthesis conditions were optimized for the PECVD growth of graphene on pristine sapphire (see Appendix A) (Table 3). The substrate was then cooled at a rate of 15 °C/min, and the samples were unloaded at temperatures below 200 °C.

### 2.4. Raman Spectroscopy

The structural quality of the synthesized graphene was characterized using Raman spectroscopy at room temperature. A confocal Raman imaging microscope (alpha 300R, WITec, Ulm, Germany) was employed in the backscattering configuration using a 100× objective and a 600 gr/mm grating with 2.8 cm^−1^ resolution. The excitation energy (wavelength) from the laser diode module was 2.33 eV (532 nm), and the power was set to 25 mW. More information about the measurement parameters can be found in the Appendix A. In order to have a representative measure of the graphene on the sapphire surface, each sample was characterized by means of 40 Raman spectra (2 accumulations, 12 s integration time) at different locations using an x-y piezo-scanner stage. The Raman spectra were then fitted using a Lorentzian function (Appendix A), extracting information on peak position, full width at half maximum (FWHM), intensity and area. The baseline correction employed was the asymmetrically reweighted penalized least squares method [30]. More information about the analysis process can be found in Appendix B.

### 2.5. Contact Angle Measurements

The needle-in-sessile-drop method was used to measure the advancing and receding contact angle of the etched sapphire substrates before and after the graphene synthesis using an automatized goniometer (90-U3-PRO, Ramé–Hart Instrument Co., Succasunna, NJ, USA) [31]. The measurements were taken at room temperature with no humidity control. Before each measurement, the samples were cleaned by the 2-solvent method described previously to avoid experimental errors caused by dirt and impurities. First, a deionized water droplet of approximately 1 mL was pumped out by a motorized micro syringe normal to the sample surface. Water was then added to the drop at very low volumes (1 µL). The contact angle in each volume step was then recorded to measure the advancing contact angle. To obtain the receding contact angle, the water of the droplet was pumped in (1 µL) until the minimum angle was measured. This advancing-receding iteration was performed 5 times, disregarding the first maximum and minimum values for the statistics because they are influenced by external factors such as airborne contamination. Afterward, the mean of the maximum (minimum) values of the advancing (receding) iteration of the 4 remaining iterations was computed.

## 3. Results and Discussion

The density of hydroxyl groups, which strongly adsorb water molecules, determines the hydrophilicity of the oxide surfaces [32]. In the case of c-plane sapphire, the Al-termination, which acts as a strong Lewis acid, promotes H_2_O adsorption [20,33]. Owing to the high reactivity exhibited by the aluminum termination upon exposure to water, the surface undergoes facile hydroxylation, leading to the generation of unbound hydroxyl groups. Consequently, the surface exhibits increased hydrophilicity, as evidenced by the enhanced wettability observed in the specimens processed using the N_2_-based process (Figure 1). In these plasma processes, the largest reduction (45%) of the contact angle (36°) is measured for the combination of 6 mbar, 100 W and 600 s as compared with the reference sample (66°).

However, the sapphire surface can also exhibit terminal oxygen bridges (Al_2_O^−^ species), which, if sufficient hydrogen is supplied, become terminal hydroxide bridges (Al_2_(OH) species) [20]. When the first layer of water molecules interacts with this surface, a small number of H atoms of water tend to form hydrogen bonds with the oxygen atoms within the terminal hydroxide bridges of the surface. Thus, a physical barrier is created, which prevents the hydroxylation of the aluminum atoms, preserving the Lewis acid sites. This process is the reason behind the increased hydrophobicity of the surface, as observed in the c-plane sapphire substrates subjected to an Ar-based plasma process (except for the combination of 4 mbar, 50 W and 300 s) [33,34]. This combination reduces the contact angle of sapphire (50°). On the other hand, the combination of 4 mbar, 100 W, 600 s and Ar; reports the largest increment (136%) of contact angle (90°), whereas the contact angles measured for the sapphire substrate etched with the combinations of 6 mbar, 100 W and 300 s and 6 mbar, 50 W, 600 s are 80° and 70° respectively. In the following, the OH-terminated sapphire substrates are considered as those that increase the hydrophobicity as compared to the reference sample.

Furthermore, the wettability properties of sapphire can be modified to a large extent using patterned nanostructures [35]. However, this effect is disregarded in our experiments, as the AFM measurements do not provide large variations of roughness after the plasma processes (see Appendix A).

When the contact angle of graphene is evaluated, a large controversy exists because of its “wetting transparency,” as the graphene layer modifies the adsorption energy between the water molecule and the substrate [36]. Contact angle measurements on free-standing graphene have shown the hydrophilic properties of graphene [37]. However, in our experiments, when the graphene is grown on the sapphire substrate, the hydrophobicity increases towards the contact angle of graphite (~90°) independently of the plasma process recipe.

### 3.1. Surface Modification of Sapphire: DoE Results

The study of the main effects (e.g., plasma power or process time) on the parameter under study (contact angle) allows us to find the influence of a particular factor within the process [29]. As has already been observed, the influence of the gas employed is remarkable; thus, it is reasonable to analyze the main effects either for the N_2_ or Ar processes (Figure 2). When the N_2_ gas is employed, the sapphire surface is modified and becomes more hydrophilic, which indicates an Al-terminated sapphire surface [20,33]. When evaluating the main effects influencing the contact angle of the sapphire surface: discharge power and pressure have a larger influence than, for example, the process time since the steeper the slope, the larger the influence of that factor.

However, when Ar is employed, the wettability of sapphire is completely different, and the surface becomes more hydrophobic, which is indicative of an OH-terminated sapphire surface [20]. In this case, our analysis shows that the pressure has a lower influence on the contact angle than the discharge power or the time of the process.

When the main effects analysis evaluates the roughness of the sapphire substrate after the plasma process (Appendix A), the parameters with a larger influence are, as expected, the process time and the discharge power (Figure 3). The discharge power provides the potential difference to accelerate the ions towards the sapphire substrate, thus transferring their kinetic energy to the surface atoms. On the other hand, the mean free path between collisions within the gas phase is determined by the process pressure. At larger pressures, the ions get thermalized as they experience more collisions in their travel toward the sapphire substrate. As observed from the main effect analysis, the surface roughness of sapphire is slightly modified by the directionality of the impinging ions towards the surface.

### 3.2. Raman Spectroscopy

The Raman fingerprint modes of pristine graphene are the G band at around 1583 cm^−1^ together with the defect-related bands: the D (1300 cm^−1^), its overtone the 2D (2680 cm^−1^) and the D′ (1620 cm^−1^) band [38]. Although the 2D band is usually referred to as the overtone of the D band, it is the most prominent feature in graphene since no defects are required for the activation of second-order phonons [39]. When the Raman spectra of the synthesized graphene are evaluated, the characteristic graphene bands fade in several Raman spectra depending on the plasma process carried out on the sapphire surface. Significantly, the first relationship between the induced terminations within the etched c-plane sapphire surfaces and the synthesis of graphene arises when evaluating the coverage of the sapphire surface. For example, full coverage is not completely achieved when the synthesis is carried out in the Al-terminated (more hydrophilic) sapphire surfaces. This, the full coverage, is studied via the data dispersion shown by the histograms of the fitted full-width-at-half-maximum (FWHM) values for the graphene D, G and 2D bands (see Appendix A). Among the Al-terminated sapphire surfaces, the full coverage is achieved for the reference sample along with one N_2_-based plasma process (4 mbar, 50 W and 600 s) and, independently of the sapphire surface termination, for every Ar-based plasma process.

The Raman fingerprint bands of graphene show different shifts depending on the sapphire termination employed (Figure 4). When graphene is grown on sapphire with an OH-termination, the graphene D and 2D bands experience a redshift above ~10 cm^−1^, whereas a blue shift (~10 cm^−1^) is experienced by the G and D′ peaks as compared with the reference sample. On the other hand, when graphene is grown on Al-terminated sapphire surfaces, no remarkable shifts are observed for the D, G, D′ and 2D band positions.

The I_D_/I_G_ ratio is usually employed to provide a measure of defects in graphene (Figure 5A). Within the Al-terminated sapphire surfaces, the I_D_/I_G_ ratios are comparable to those of the reference sample (3.48). However, the I_D_/I_G_ ratio decreases for graphene grown on OH-terminated sapphire surfaces where a reduction of 38% is observed (Ar, 4 mbar, 100 W and 600 s).

On the other hand, the structural quality is usually evaluated by the I_2D_/I_G_ ratio. Independently of either the Al- or OH- terminated sapphire surface, the I_2D_/I_G_ ratio decreases as compared with the reference sample (I_2D_/I_G_ ratio ~0.37). Both smaller (30%) and larger (59%) reductions are observed for the graphene grown on OH-terminated sapphire surfaces.

The presence of sharp defect-related D and D′ bands, together with the presence of the D + D′ band and the position of the G band, allows us to identify the synthesized graphene within stage I of the amorphization trajectory in graphite [41]. In this stage, the D-band scattering is proportional to the average distance between nearest defects (L_D_) or the defect density (σ); thus, $I_{D}/I_{G}\propto \;1/L_{D}^{2}\;\propto \;\sigma ,$ which can be calculated in terms of the laser excitation wavelength λ_L_ (nm) after Equations (1) and Equation (2), respectively [42].
(1)$$L_{D}^{2}\;\left({\mathrm{nm}}^{2}\right)=1.8\times 10^{-9}\lambda_{L}^{4}{\left(\frac{I_{D}}{I_{G}}\right)}^{-1}$$
(2)$$\sigma \;\left({\mathrm{nm}}^{-2}\right)=7.3\times 10^{-9}\lambda_{L}^{4}\left(\frac{I_{D}}{I_{G}}\right)$$

Therefore the I_D_/I_D′_ ratio does not depend on the defect concentration but on the nature of the defects [43]. This ratio has been associated with boundary-like defects for values close to 3.5 and to vacancy-like defects when I_D_/I_D′_ tends to seven. Within our samples, there are two different regimes depending on the plasma-induced sapphire surface terminations (Figure 5B). The graphene samples grown on the Al-terminated sapphire surfaces reports an I_D_/I_D′_ ratio, which tends towards the vacancy-like regime. On the other hand, the OH-terminated surfaces tend toward a boundary-like regime.

Furthermore, the contributions of point and line defects in the Raman spectra can be further investigated from the I_D_/I_G_ ratio [44]. The point defects are considered 0D defects, and they are characterized by the average distance between nearest defects (L_D_). For a perfect graphene $L_{D}\to \infty$, whereas for a fully disordered graphene L_D_ → 0. On the other hand, line defects (1D) are evaluated by the crystallite size (L_a_) Equation (3) or by the crystallite area ($L_{a}^{2}$) [45].
(3)$$L_{a}\;\left(\mathrm{nm}\right)=\left(2.4\times 10^{-10}\right)\lambda_{L}^{4}{\left(\frac{I_{D}}{I_{G}}\right)}^{-1}$$

Similar to the analysis method proposed by Cançado et al. [44], the effect of the plasma-induced surface terminations is evidenced in the variations of the defect density σ and crystallite area $L_{a}^{2}$ (Figure 6). Interestingly, the Al-terminated sapphire surfaces correspond to a larger number of defects and smaller crystallite areas. On the other hand, OH-terminated surfaces have smaller defect densities and larger crystallite areas.

The former is explained by the catalytic effect of the superficial aluminum. At temperatures above 500 °C, the hydroxyls of the surface are removed, leaving highly aluminum-rich surfaces, thus favoring the catalytic effect of Al during the synthesis [32,46]. There are several reports where the growth rates of graphene on different sapphire planes are evaluated, and interestingly, the catalytic effect of aluminum on the surface is also reported [23,24]. According to Ueda et al., the catalytic effect of the r-plane sapphire surface promotes the full coverage of the sapphire surface [24,47]. However, their CVD process is carried out at 1200 °C. In the case of c-plane sapphire, both Saito et al. and Ueda et al. reported that the growth of graphene is only observed in the pits of Al-rich surfaces caused by the thermal desorption of oxygen atoms within the surface. On the other hand, the synthesis route proposed in this work controls the surface termination of the c-plane sapphire using a plasma process; furthermore, the PECVD technique, which reduces the temperature needed for the precursor to dissociate, allows the synthesis at lower temperatures.

## 4. Conclusions

In conclusion, our experiments show that the plasma-induced modifications of the c-plane sapphire surface play a role in the PECVD synthesis of graphene at 800 °C. By controlling the plasma process through a combination of process pressure, discharge power, time and type of gas, the sapphire surface termination can be engineered. In this work, either Al- or OH- terminations of the sapphire surface have been identified by contact angle measurements. As compared with the pristine c-plane sapphire surface, the OH-terminated surface increases the hydrophobicity of the sapphire surface, whereas an Al-terminated surface reduces its contact angle with the water droplet, thus being more hydrophilic. The AFM analysis demonstrates that the wettability behavior is not due to the surface roughness created by any etching process induced by the plasma process but to a different chemical termination of the surface.

The quality of the PECVD synthesized graphene on the plasma etched sapphire substrates is studied by its fingerprint Raman resonances, namely its D, G, D′ and 2D bands. The full coverage of the sapphire surface is achieved for all OH-terminated surfaces. On the contrary, not every Al-terminated surface shows full coverage, understood as a small dispersion of the fitted FWHM values for the D, G and 2D bands.

The I_D_/I_G_ ratio, usually employed as a measure of defectiveness in graphene, is substantially reduced when graphene is synthesized on the OH-terminated surfaces (Ar-based process). On the other hand, independently of the plasma process, the I_2D_/I_G_ ratio, which is usually reported as a quality measure of graphene, is reduced as compared with the pristine c-plane sapphire. The nature of graphene defects was identified by the I_D_/I_D′_ ratio: Al-terminated sapphire surfaces show vacancy-like defects in graphene, whereas boundary-like defects are more prominent in the graphene grown on OH-terminated sapphire surfaces.

The representation of defect density ($\sigma =1/L_{D}^{2}$) versus crystallite area ($L_{a}^{2}$) allows us to discern how the graphene crystallite size and defectiveness are largely influenced by the plasma process carried out on the sapphire substrate. These results provide insight into the synthesis process of graphene on c-plane sapphire and ways to improve the quality of graphene without the need for changing the sapphire crystal orientation.

## Figures and Tables

**Figure 1 nanomaterials-13-01952-f001:**
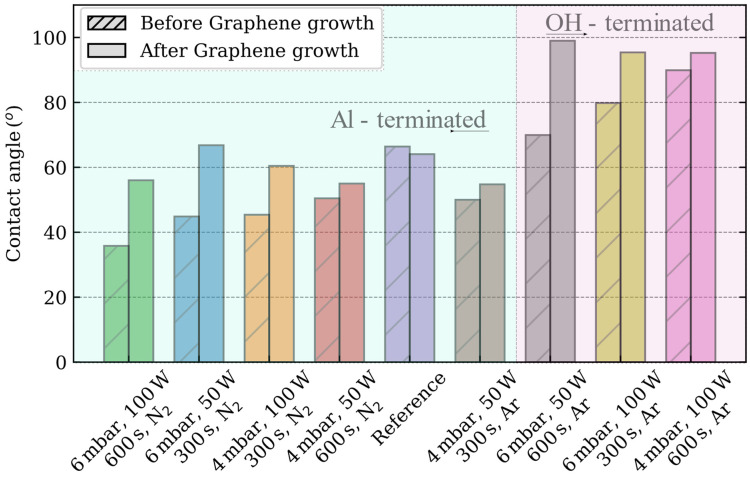
Advancing contact angle measurements of the c-plane sapphire substrates depends on the plasma process carried out.

**Figure 2 nanomaterials-13-01952-f002:**
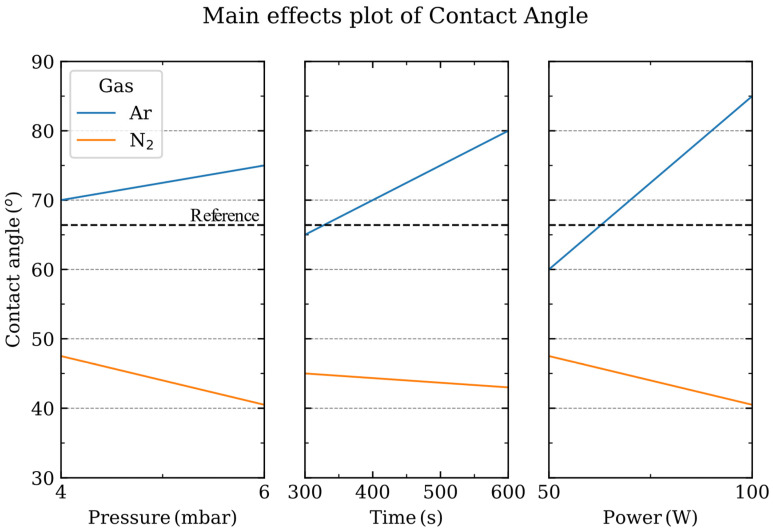
Main effects plot of pressure, power and time depending on the gas employed during the plasma process. The contact angle of the pristine sapphire substrate is shown in the dashed line.

**Figure 3 nanomaterials-13-01952-f003:**
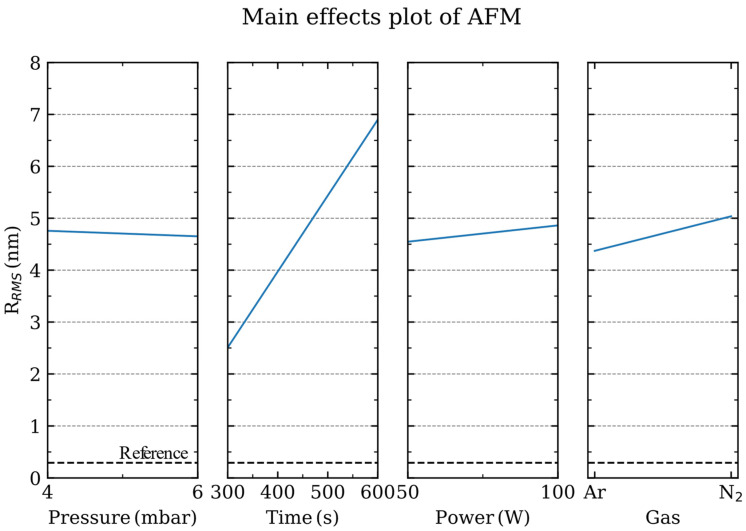
Main effects plot of the root mean square roughness (R_RMS_) for the pressure, power, time and gas employed during the plasma process. The R_RMS_ value of the pristine sapphire substrate is shown in the dashed line.

**Figure 4 nanomaterials-13-01952-f004:**
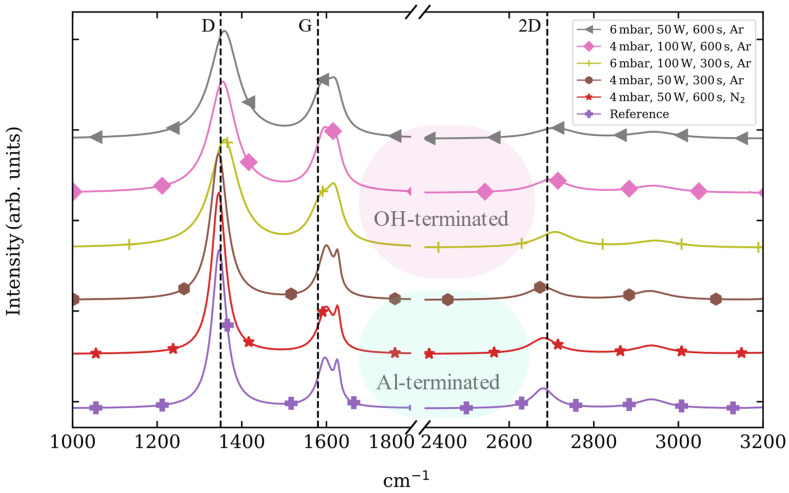
Standard normal variate normalization of the graphene Raman spectra grown by PECVD on sapphire substrates etched with different combinations of pressure, power, time and gas [40]. The lines are shifted in intensity, and vertical lines showing the position of D, G and 2D graphene bands are included for the aid of visualization.

**Figure 5 nanomaterials-13-01952-f005:**
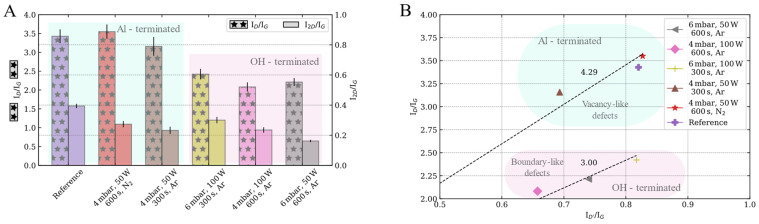
Intensity ratios of the characteristic graphene Raman bands of the samples synthesized on sapphire substrates etched with different combinations of pressure, power, time and gas. (**A**): I_D_/I_G_ and I_2D_/I_G_ ratios. (**B**): I_D_/I_G_ vs. I_D′_/I_G_ ratios, which can be employed to evaluate the nature of defects.

**Figure 6 nanomaterials-13-01952-f006:**
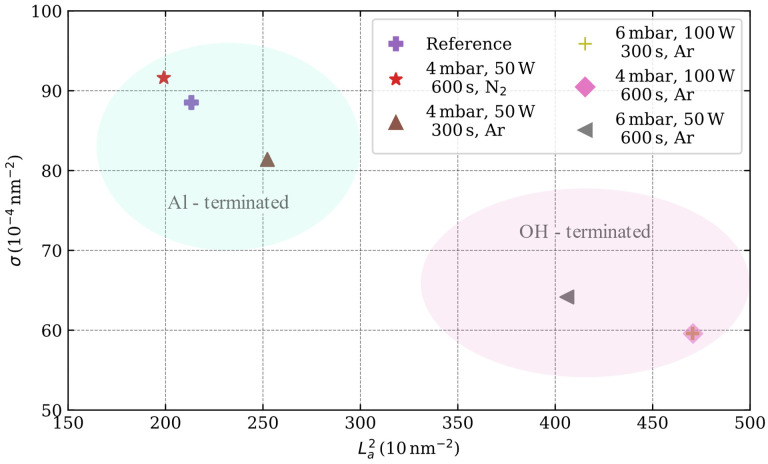
Defect density ($\sigma =1/L_{D}^{2}$) versus crystallite area ($L_{a}^{2}$) with the L_a_ and L_D_ values as calculated from Equation (2) and Equation (3), respectively.

**Table 1 nanomaterials-13-01952-t001:** Comparison of different growth technologies and parameters employed in the synthesis of graphene on sapphire substrates.

Reference	Substrate	Catalyst	Technology	Plasma Power (W)	Temperature (°C)	Pressure (mbar)	Gas Ratio (Ar:H_2_:CH_4_)
[1] N. Mishra et al.	c-Sapphire	-	CVD	-	1200	25	200:20:1
[6] H. J. Song et al.	c-Sapphire	-	CVD	-	950	Atmospheric	0:5:3
[10] D. Wei et al.	c-Sapphire and SiO_2_/Si	-	PECVD	80	500–700	64	0:3:10
[23] K. Saito	a- and c- Sapphire	-	CVD	-	1000	Atmospheric	10:5:10
[24] Y. Ueda et al.	a-, c- and r- Sapphire	-	LPCVD	-	1090–1210	100	N_2_:H_2_:Bubbled-C_6_H_10_
[25] J. Li et al.	c-Sapphire	Cu(111)	CVD	-	1075	4	35:5:1 and 5:1:1(diluted CH_4_)
This work	c-Sapphire	-	PECVD	100	800	4	10 N_2_:15:1

**Table 2 nanomaterials-13-01952-t002:** Table of signs employed for the 2^4−1^ fractional factorial design. Signs are in brackets as a guide to understanding the creation of the L_8_ orthogonal array.

	Factors			
	a	b	c	abc
Run Number	Pressure [mbar]	Power [W]	Time [s]	Gas Type
1	4 (−)	100 (+)	600 (+)	Ar (−)
2	6 (+)	100 (+)	600 (+)	N_2_ (+)
3	6 (+)	50 (−)	300 (−)	N_2_ (+)
4	4 (−)	50 (−)	300 (−)	Ar (−)
5	6 (+)	100 (+)	300 (−)	Ar (−)
6	4 (−)	50 (−)	600 (+)	N_2_ (+)
7	6 (+)	50 (−)	600 (+)	Ar (−)
8	4 (−)	100 (+)	300 (−)	N_2_ (+)

**Table 3 nanomaterials-13-01952-t003:** Process parameters for the synthesis of graphene on the etched sapphire substrate.

Temperature [°C]	Pressure [mbar]	Plasma Power [W]	N2 [sccm]	Ar [sccm]	CH_4_ [sccm]	H_2_ [sccm]	Time [s]
800	4	100	100	0	20	300	1200

## Data Availability

Data will be provided upon reasonable request to the corresponding author.

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
