# Peer review of "Plasma-Induced Surface Modification of Sapphire and Its Influence on Graphene Grown by Plasma-Enhanced Chemical Vapour Deposition"

_nanomaterials, 2023, doi:10.3390/nano13131952_

Round 1
Reviewer 1 Report
In this work, the authors investigated the effect of different surface terminations on the synthesis of graphene by plasma enhanced chemical vapor deposition on c-sided sapphire substrates. The different terminations of the sapphire surface are controlled by a plasma process. A design of experiments procedure was carried out to evaluate the major effects governing the plasma process of four different parameters: i.e., discharge power, time, pressure and gas employed. I believe that publication of the manuscript may be considered only after the following issues have been resolved.
1. In order to better highlight the advantages of this work, the author needs to provide a table to compare related work.
2. There have been many reports on the growth of graphene by chemical vapor deposition. What are the advantages of this work? Suggest the author to provide an exploration of relevant growth mechanisms.
3. The introduction can be improved. The articles related to some applications of graphene and graphene oxide materials should be added such as Results in Physics 48, 2023, 106420; Micromachines 2023, 14, 953; Commun. Theor. Phys. 2023, 75, 045503; Optics Express, 30(20), 35554-35566, 2022.
4. Please check the grammar and spelling mistakes of the whole manuscript.
Minor editing of English language required
Reviewer 2 Report
The work of Lozano et al. regarding the surface modification of sapphire via plasma treatment for the graphene growth via plasma-enhanced chemical vapor deposition is well written and assess a relevant topic in the field of wafer-scale synthesis of graphene.
The work is well made and the characterization of the material (both growing substrate sapphire and growth graphene) is adequate. The paper is well written and clear. I just have few comments, that in my opinion could improve the quality of the work:
1- In the methods there should be a more detailed presentation of the characterization techniques (i.e. the contact angle and AFM parameters used).
2- The Raman acquisition time is quite high compared with the usual parameters founds in literature, the same can be said for the laser power. Can the authors discuss on that? Why were those parameters necessary so high?
3- In figures 2 and 3 the authors should add, as a reference, the values for the untreated substrate.
4- In figure 4 the authors could add a guideline for the expected D,G and 2D peak position, to help the readers to better understand the shifts.
Overall, I believe this work suitable for the publication in nanomaterials, after these minor corrections.
Round 2
Reviewer 1 Report
Accept.